# Post-vaccination *Streptococcus pneumoniae* colonization and respiratory manifestations in children: A prospective cohort study

**Jaqueline Elisa Verardo Benedetti**[1⊙], **Kauana Pizzutti**[1‡], **Mariana Preussler Mott**[1‡], **Pedro Uriel Pedrotti Vieira**[2,‡], **Neide Maria Bruscato**[3,‡], **Emilio Hideyuki Moriguchi**[3,4], **Roberta Rigo Dalla Corte**[4], **João Carlos Batista Santana**[3,5,6⊙], **Cícero Dias**[1,2⊙¤*]

**1** Departamento de Epidemiologia e Métodos Diagnósticos, Programa de Pós Graduação Ciências da Saúde, Universidade Federal de Ciências da Saúde de Porto Alegre - UFCSPA, Porto Alegre, Rio Grande do Sul, Brazil, **2** Departamento de Ciências Básicas da Saúde, Universidade Federal de Ciências da Saúde de Porto Alegre - UFCSPA, Porto Alegre, Rio Grande do Sul, Brazil, **3** Departamento de Pesquisa, Instituto Moriguchi, Veranópolis, Rio Grande do Sul, Brazil, **4** Departamento de Medicina Interna, Universidade Federal do Rio Grande do Sul - UFRGS, Porto Alegre, Rio Grande do Sul, Brazil, **5** Departamento de Pediatria, Universidade Federal do Rio Grande do Sul - UFRGS, Porto Alegre, Rio Grande do Sul, Brazil, **6** Departamento de Pediatria, Universidade do Vale dos Sinos - Unisinos, São Leopoldo, Rio Grande do Sul, Brazil

⊙ These authors contributed equally to this work.
‡ KP, MPM and NMB also contributed equally to this work.
¤ Current address: Departamento de Ciências Básicas da Saúde, Universidade Federal de Ciências da Saúde de Porto Alegre – UFCSPA, Porto Alegre, Rio Grande do Sul, Brazil
* cicerodias17@gmail.com, cicero@ufcspa.edu.br

## Abstract

### Background

*Streptococcus pneumoniae* is considered one of the main agents for pneumonia, meningitis, bacteremia, sinusitis, and acute otitis media (AOM)especially in children under 5 years, and a cause for morbidity and mortality due to respiratory infections worldwide. Our aim was to investigate the influence of *Streptococcus pneumoniae* colonization in vaccinated children regarding infections in a one-year follow-up.

### Methods

A double-blind, observational, prospective cohort study was conducted on children aged 18–59 months, vaccinated with pneumococcal conjugate vaccine 10 (PCV10) or pneumococcal conjugate vaccine 13 (PCV13). A total of 225 children were monitored, with different dates of entry into the study, which occurred between March 2018 and October 2019 (zero time). At the end of one year, counting from the date of entry, interviews and data collection took place in medical records (end of follow-up). The Poisson regression with robust variance and Chi-squared or Fisher's exact tests were used for qualitative analyses; Mann-Whitney or Friedman tests for quantitative analyses.

**Data availability statement:** All relevant data are within the manuscript. Database available on request to the author.

**Funding:** The author(s) received no specific funding for this work.

**Competing interests:** The authors have declared that no competing interests exist.

## Results

A high colonization rate (64.4%) was observed, with only 2.8% of carriers having a PCV10 vaccine serotype, specifically 6B, as expected. Being male showed association to colonization ($p = 0.05$). We found that children colonized by pneumococcus do not have an increased risk for respiratory diseases or antimicrobial use. Exception was only observed in cases of serotype 6B colonization, showing association with pneumonia in children under 2 years ($p = 0.016$).

## Conclusion

Our study reveals that the carriage of *Streptococcus pneumoniae* does not appear to significantly impact the incidence of respiratory diseases in a fully-vaccinated children population. However, it is noteworthy that a correlation was observed in the occurrence of pneumonia in children under the age of 2 when colonized by serotype 6B.

## Introduction

*Streptococcus pneumoniae* is considered one of the main agents for pneumonia, meningitis, bacteremia, sinusitis, and acute otitis media (AOM) [1,2], especially in children under 5 years, and a cause for morbidity and mortality due to respiratory infections worldwide [3]. In their study, Wahl *et al*. (2018) estimated that there were 294,000 pneumococcal deaths among children aged 1–59 months worldwide in 2015 and that these deaths decreased by 51% from 2000 to 2015 due to the widespread use of PCV accompanied by also a reduction in cases of the disease [4,5]. The same study highlighted that Brazil reached one of the largest reductions by 2015, 88% of deaths from this agent after the introduction of the 10-valent pneumococcal conjugate vaccine (PCV10) in the country [4]. More recently, in a hospital-based study historical series on IPD among children in Brazil, fatalities were significantly reduced from 6.6 to 2.0 cases per 10,000 after the introduction of PCV10 in the public system [6]. In the state of Rio Grande do Sul, a decrease in the incidence of meningitis has been observed over time [7].

Prevention for pneumococcal disease is based mainly on active immunization [8], with the capsule being its main target, in which polysaccharide antigens induce serotype-specific response [9,10]. A study was conducted with Brazilian children in 3 moments (2010, 2013 and 2017), and saw that the high prevalence of PCV10 types (47.8%) before vaccination decreased in 2017 to a few isolates (1.4%), represented by serotypes 6B, 9V, 14, 18C and 23F, showing a robust impact of vaccination [11]. The same study showed, through laboratory-base surveillance for invasive pneumococcal disease (IPD) pre- and post-PCV10 introduction, that the occurrence of cases involving non-vaccine serotypes (NVT) increased from 20.7% to 88.7%, with serotypes 3, 6C and 19A being the most prevalent [11]. In a time-trend analysis conducted in high-income countries, NVT were observed to be the most prevalent in post-PCV era, but no dominant serotype was observed [12]. Post-conjugate vaccines

introduction studies did not find decrease in overall carriage rates in the population, but rather replacement, where non-vaccine serotypes (NVT) have become more prevalent [13–15].

Clinical expression of *S. pneumoniae* in the colonization/disease binomial can occur via non-invasive strains with long time of colonization and low risk of tissue invasion, or via specifically selected clones whose capacity for quick induction of the disease [16] leads to IPD [17]. *S. pneumoniae* has a remarkable ability of avoiding or taking advantage of the host's inflammatory and immune responses, which allows it to invade normally sterile sites, such as spaces in middle ear, lungs, bloodstream and meninges [1,9].

There are few longitudinal and prospective studies relating colonization and respiratory disease by monitoring children. Thus, we aimed to investigate the consequences of colonization by *Streptococcus pneumoniae* in terms of respiratory infections in a cohort of Brazilian children, during a one-year follow-up.

## Methods and material

### Study design

This was a prospective, double-blind, follow-up, observational cohort study with children aged 18–59 months undertaken in Veranópolis – RS, Brazil from Feb/2018 to Oct/2019. Each child was followed up for up to one year after their entry into the study (Feb/2019 to Oct 2020).

### Profile of the study area

This town had an estimated population of 26,533 (87% urban) people in 2020, of which 781 were children aged 18–59 months. Its 0.773 Human Development Index (HDI), U$ 15,400.00 Gross Domestic Product (GDP) (IBGE, 2018), and 75.3-year-old life expectancy at birth (IBGE, 2000) [18]. The care network has 15 pediatric health services (7 public, including 1 hospital, and 8 private). In the period 2018–2022 there was 1 death in the age group of 1–4 years in the city (coefficient 3.8/100,000) show that this city has good lifestyle indicators.

This study is reported following the STROBE guidelines [19].

### Recruitment

This trial was part of a larger ongoing study called "*Streptococcus pneumoniae* carriers among elderly and children: evaluation of the effect of PCV10", initiated in 2018, in which 231 children aged 18–59 months participated.

The trial used the same cohort of 231 children show in Fig 1. Eligibility required children aged 18–59 months, enrolled in it, and having specimens collected from their nasopharynx (NP). Study entry occurred between March 2018 and October 2019 (zero time), with follow-up interviews and medical record reviews conducted one year later. In cases of refusal to have the NP swab collected or to answer the proposed questionnaire a year later, the participant was excluded from the study.

A participant recruitment strategy was established. Following the random draw order, 11 out of 13 school principals were contacted to ensure a randomized selection. School visits continued until the required sample size was reached, considering a loss of 10%, resulting in different entry and exit dates for each participant. Additional volunteers outside the school environment were recruited through an appeal on local radio.

### Follow-up

The following time references were used: "zero time" — beginning of observation, date of the NP swab collection; "end of follow-up" — 12 months after "zero time". An interview was conducted at "end of follow-up" using a structured questionnaire with closed-ended questions to obtain primary data about sociodemographic characteristics, clinical conditions (number and type of event related to respiratory disease), hospitalization, use of antibiotics, vaccination and exposure

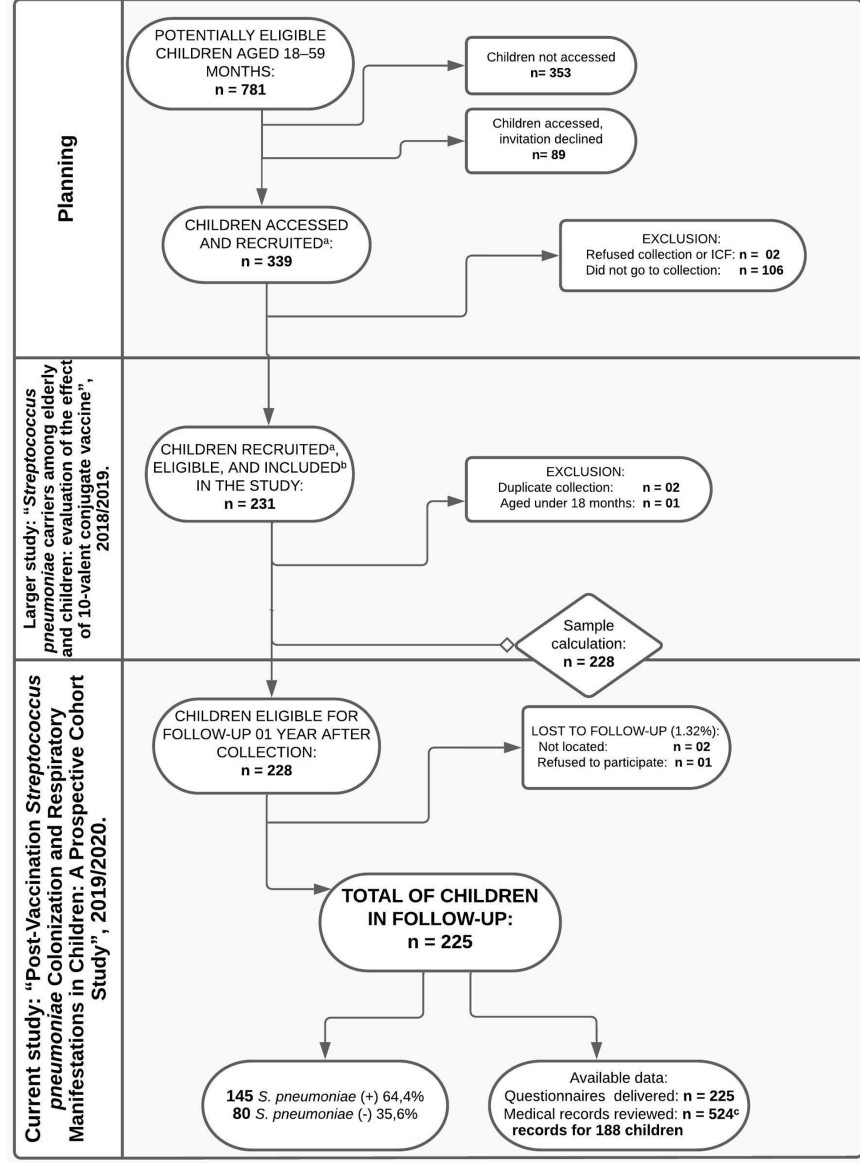

**Fig 1. Flowchart.** [a] Recruiting via: phone call, letter to the parents, community health workers, in-person contact with the parents at school or vaccination campaign, invitation on social media. [b] Inclusion criteria: child aged 18–59 months with specimens collected from their NP, sample considered adequate, and having participated in the previous study. [c] May have 1 or more records in different health services.

factors. Each participant's vaccination card was used to obtain data on which vaccine was used (PCV10 or PCV13) and when it was administered. Vaccination schedules were analyzed, and children were considered fully vaccinated with the following criteria, according to the National Immunization Program: if vaccinated with PCV10 before 2016, three doses and a booster; as of 2016, two shots and a booster; PCV13, three shots and a booster. We also considered the possibility of mixed regimens (PCV10 + PCV13) [20,21].

Besides the interview, data from each participant's medical record were collected (date and place of the consultation, use of antimicrobials, signs and symptoms). In the local hospital, a search was made by records of children who have

been hospitalized, especially with a diagnosis of pneumonia. They were accessed through 15 different health services to search for relevant outcomes for the study. These outcomes were classified according to the clinical syndrome: community-acquired pneumonia (CAP), sinusitis, AOM, tonsillopharyngitis, asthma and upper and lower Acute Respiratory Infection (ARI).

All outcomes of interest were registered in the database, but the occurrence of each one of them was computed only once per child, even if the same outcome occurred twice for the same participant. If there was no record of visits for a child, we assumed that they did not receive medical care for the outcomes of interest in that period. The diagnoses obtained from the medical records were based exclusively on the criteria of each physician responsible for the child's care. There was no training aimed at standardizing criteria among professionals. For the diagnosis of pneumonia, we adopted criteria used by Shah *at al.*, 2017: children with cough and/or difficulty breathing, with or without fever in the presence of rapid breathing or subcostal retraction [22,23]. All pneumonia cases were reviewed and confirmed/ruled by our team's pediatrician (JCBS), considering the previously described criteria for case entry together with the demonstration of infiltrate or consolidation on chest X-ray [24]. A single researcher (JEV) conducted the interviews and the data collection from the records. It should be noted that NP collection occurred in one moment only, at "zero time", and data for respiratory disease were collected through the one-year observation period. Two different databases were generated using Epi Info™ 7 version 7.2.4.0 for the entry of data from the interviews and the records.

## Sample collection and laboratory procedures

For each eligible individual, the NP swab was collected and immediately put in cryotubes with 1.0 ml of transport medium (STGG) in which it was stored and conserved in a biofreezer at −80ºC until its processing. All samples were considered adequate and remained viable until their analysis. The first stage took place in the Microbiology Laboratory at the Federal University of Health Sciences of Porto Alegre (UFCSPA), where culture and optochin susceptibility and bile solubility tests were conducted. In the same lab, the method used for serotype deduction was multiplex PCR [25]. Then the samples were sent to the Centers for Disease Control and Prevention (CDC) for Quellung reaction.

## Sample size

The sample was calculated to find a colonization prevalence of 62.3% (Brandileone *et al.*, 2019) [11], with 95% confidence, acceptable error of 5.3% and a population of 781 children aged 18–59 months from Veranópolis, which resulted in 228 children.

## Statistical analysis

Results for qualitative variables were presented as frequency and percentage. In the records, the existence of each diagnosis was defined as the presence of the pathology in at least one visit (including sampled children who did not have visits). The overall number of visits, with each diagnosis and season of the year, was presented as mean, standard deviation and median. The association between qualitative variables was verified through the Chi-squared test or the Fisher's exact test, when appropriate, and the estimates for relative risk (RR) with CI95% were obtained through Poisson regression analysis with robust variance. The comparison of number of visits according to the colonization was obtained through the Mann-Whitney test. The Friedman test with the Dunn's test for multiple comparisons was used to compare if there was a difference in the number of visits according to the season. Results were considered significant with p-value<0.05, and were analyzed with SPSS (IBM SPSS Statistics for Windows, Version 25.0. Armonk, NY: IBM Corp.).

## Ethical consideration

All legal guardians provided written informed consent signed at the moment of the collection through Informed Consent Form (ICF). This study was approved by UFCSPA and HCPA (Clinics Hospital of Porto Alegre) ethics committees

(#2.176.785 and #2.106.235, respectively; #3.063.051 and #3.374.087 amendments for the latter), and was conducted in accordance with the Declaration of Helsinki principles (June/1964, latest amendment in 2000).

## Results

Among the 228 participants eligible for follow-up, there was a loss rate of 1.32% (Fig 1) yielding 225 children were monitored in 2019/2020 ("end follow-up"). Each child had unique entry and exit dates, and one year after "zero time" 216 phone interviews and 09 in-person interviews were conducted.

A 64.4% (n = 145/225) colonization rate was found, while in 35.6% (n = 80/225) of the children the agent was not found.

Table 1 depicts demographic characteristics for colonized and non-colonized groups regarding age, sex, living conditions, living with a smoker, smoke exposure and vaccination status. We observed higher proportion of colonization among male children — 81 of them (p = 0.05), which presented a 21% increase in risk of colonization (CI95% 0.99–1.48) when compared to female participants (n = 64/110).

**Table 1. Cohort of children aged 18–59 months, colonized and non-colonized: general (socioeconomic/demographic) and vaccination characteristics, Veranópolis, Brazil — Interview database, n = 225.**

| VARIABLE | S. pneumoniae (-) n=80 n (%) | S. pneumoniae (+) n=145 n (%) | Univariate p | RR | CI95% | Multivariate p | RR | CI95% |
|---|---|---|---|---|---|---|---|---|
| **Age: 18–23 months** | 16 (37.2%) | 27 (62.8%) | | 1 | | | 1 | |
| 24–35 months | 23 (36.5%) | 40 (63.5%) | 0.941 | 1.011 | 0.752-1.360 | 0.840 | 1.030 | 0.773-1.372 |
| 36–47 months | 21 (33.9%) | 41 (66.1%) | 0.727 | 1.053 | 0.787-1.409 | 0.522 | 1.104 | 0.815-1.497 |
| 48–59 months | 20 (35.1%) | 37 (64.9%) | 0.828 | 1.034 | 0.767-1.394 | 0.463 | 1.118 | 0.830-1.506 |
| **Sex: Female** | 46 (41.8%) | 64 (58.2%) | | 1 | | | 1 | |
| Male | 34 (29.6%) | 81 (70.4%) | **0.05** | 1.21 | 0.99-1.48 | 0.117 | 1.177 | 0.960-1.443 |
| **Household[a]: 2–3 people** | 33 (35.1%) | 61 (64.9%) | | 1 | | | 1 | |
| 4–5 people | 40 (34.8%) | 75 (65.2%) | 0.961 | 1.005 | 0.823-1.227 | 0.585 | 1.061 | 0.857-1.314 |
| 6 or more | 7 (43.8%) | 9 (56.3%) | 0.540 | 0.867 | 0.549-1.369 | 0.404 | 08.26 | 0.526-1.296 |
| **Children in the household[b]: 01** | 64 (33.9%) | 125 (66.1%) | | 1 | | | 1 | |
| 2–3 | 16 (44.4%) | 20 (55.6%) | 0.269 | 1.190 | 0.874-1.622 | 0.194 | 0.802 | 0.574-1.119 |
| **Shared bedroom[c]: No** | 45 (34.1%) | 87 (65.9%) | | 1 | | | 1 | |
| Yes | 35 (37.6%) | 58 (62.4%) | 0.588 | 0.946 | 0.775-1.156 | 0.363 | 0.906 | 0.732-1.121 |
| **Smoker family member: No** | 64 (35%) | 119 (65%) | | 1 | | | 1 | |
| Yes | 16 (38.1%) | 26 (61.9%) | 0.703 | 0.952 | 0.734-1.235 | 0.529 | 0.924 | 0.723-1.181 |
| **Smoke exposure[d]: No** | 36 (37.1%) | 61 (62.9%) | | 1 | | | 1 | |
| Yes | 44 (34.4%) | 84 (65.6%) | 0.671 | 1.044 | 0.856-1.272 | 0.870 | 1.017 | 0.834-1.239 |
| **School[e]: No** | 2 (40%) | 3 (60%) | | 1 | | | 1 | |
| Yes | 78 (35.5%) | 142 (64.5%) | 0.843 | 1.076 | 0.522-2.215 | 0.729 | 1.128 | 0.570-2.234 |
| **Vaccination: Mixed[f]** | 6 (42.9%) | 8 (57.1%) | | 1 | | | 1 | |
| PCV13[g] | 3 (50%) | 3 (50%) | 0.776 | 0.875 | 0.349-2.195 | 0.796 | 0.887 | 0.358-2.199 |
| PCV10[h] | 71 (34.6%) | 134 (65.4%) | 0.570 | 1.144 | 0.719-1.820 | 0.599 | 1.132 | 0.713-1.796 |

*S. pneumoniae* (-) = non-colonized, *S. pneumoniae* (+) = colonized. [a] Number of people living in the same household (including the child). [b] Number of children under 5 in the household (including the child). [c] Child shares the bedroom or bed with at least one person. [d] Use of fireplace or wood cookstove in the house. [e] School attendance between "zero time" and "end of follow-up". [f] Mixed vaccination — the child received PCV10 and PCV13 doses. [g] Received 4 doses of PCV13. [h] Received 3 or 4 doses of PCV10.

All children (100%) enrolled in the cohort were considered vaccinated for their age and the period in question, since there was a change in sequential schedule for doses in Brazil, from 3 + 1–2 + 1, in 2016. A total of 205 (91.11%) children (134 colonized and 71 non-colonized) received 3 or 4 doses of PCV10, 6 (2.67%) received 4 doses of PCV13 (3 colonized and 3 non-colonized) and in 14 children (6.22%) vaccines were interchanged (mixed schedule, PCV10 + PCV13), among which there were 8 colonized and 6 non-colonized. There was no association between the vaccine received and colonization, regardless of the product used.

For 188 participants, 524 records were accessed, and 765 medical visits were generated (with one or more outcomes of interest recorded) between "zero time" and "end of follow-up", with visits ranging from 1 to 15 per participant. For 37 (16.44%) of the 225 children followed, no medical care was registered in the period in question, among which 78.4% (n = 29/37) were colonized and 21.6% (8/37) were non-colonized. The mean of the number of visits for colonized and non-colonized was the same, 3.4 visits/child/period. Among the 145 colonized, 80% had at least one visit in the period (n = 116/145). Among the non-colonized, 90% had at least one visit recorded in the period (72/80). However, the difference was not significant (p = 0.053, CI95%, 0.80–0.99). Regarding seasonality, visits mean was also very similar among colonized and non-colonized. The seasons that showed higher mean of visits were winter (1.1 visit for colonized and 1 for non-colonized) and spring (1.1 visit for both groups).

For both carriers and non-carriers of *S. pneumoniae*, observed rates for pneumonia, AOM, sinusitis, upper and lower tract ARI, tonsillopharyngitis and asthma were very similar, with no association between pathology and colonization verified, as shown on Table 2. For pneumonia, sinusitis, and AOM, the mean of visits among colonized and non-colonized was 0.05 versus 0.03, 0.2 for both, and 0.2 and 0.3, respectively. On average, in 51.6% (n = 116/225) of medical care visits for *S. pneumonia* carriers and non-carriers, antimicrobials were prescribed (51.7%, n = 75/145 versus 51.3%, n = 41/80).

A total of 19 different serotypes were identified and occurred in 147 isolates (146 with defined serotype and 2 isolates defined as nontypeable) from 145 positive children (2 participants were identified as simultaneous carriers — more than one serotype). A PCV10 serotype (serotype 6B) was found in 4/145 (2.8%) children, and colonization by an additional

**Table 2. Distribution of outcomes per clinic syndrome[a] among children aged 18–59 months with medical care in record, Veranópolis, Brazil, n = 225.**

| | *S.pneumoniae* (-) n = 80 | | *S.pneumoniae* (+) n = 145 | | Univariate | | | Multivariate | | |
|---|---|---|---|---|---|---|---|---|---|---|
| | n | % | n | % | p | RR | IC95% | p | RR | IC95% |
| Pneumonia | 2 | 2.50 | 5 | 3.4 | 0.695 | 1.38 | 0.27-6.95 | 0.714 | 1.37 | 0.25-7.36 |
| Sinusitis | 14 | 17.50 | 25 | 17.2 | 0.961 | 0.99 | 0.54-1.79 | 0.985 | 1.01 | 0.55-1.84 |
| AOM[b] | 14 | 17.50 | 26 | 17.9 | 0.935 | 1.02 | 0.57-1.85 | 0.850 | 1.06 | 0.59-1.90 |
| Tonsillopharyngitis | 22 | 27.50 | 36 | 24.8 | 0.661 | 0.9 | 0.57-1.42 | 0.548 | 0.87 | 0.55-1.37 |
| ARI[c] | 66 | 82.50 | 109 | 75.2 | 0.206 | 0.91 | 0.79-1.05 | 0.382 | 0.94 | 0.82-1.08 |
| Asthma | 2 | 2.50 | 9 | 6.2 | 0.217 | 2.48 | 0.55-11.2 | 0.319 | 2.22 | 0.46-0.72 |
| Antimicrobial[d] | 41 | 51.30 | 75 | 51.7 | 0.946 | 1.01 | 0.77-1.32 | 0.972 | 1.00 | 0.77-1.31 |
| No medical care[e] | 8 | 10 | 29 | 20 | 0.064 | 2.00 | 0.96-4.17 | 0.096 | 1.82 | 0.90-3.68 |

[a] Presence of at least one episode of respiratory disease (pneumonia, sinusitis, AOM, ARI, asthma or tonsillopharyngitis) between "zero time" and "end of follow-up". [b] AOM — acute otitis media. [c] ARI — Upper and lower respiratory tract infection — diagnosis classified in this category when record showed: cold, flu, rhinitis, allergic/viral rhinitis, respiratory virosis, conjunctivitis, influenza, viral tonsillitis, sore throat, nasopharyngitis, pharyngitis, laryngitis, turbinate hypertrophy, enlarged adenoids, tracheobronchitis, bronchitis, bronchiolitis, acute respiratory infection (ARI), head, foot, and mouth disease, erythema multiform, mononucleosis, adenovirus infection, unspecified viral infection, viral/external/serous AOM, positive COVID test. [d] Record showed one or more uses of antimicrobial between "zero time" and "end of follow-up". [e] Children without medical care record in the period of interest.

PCV13 serotype (3, 6A or 19A) was found in 58/145 (40.0%) children. Most carrier children were colonized by non-PCV10/PCV13 serotypes: 57.2% (n = 83/145) of the total.

Data show that 3.1% (7/225) of children had community-acquired pneumonia (CAP) in the period, among which 2 were non-colonized and 5 were *S. pneumoniae* carriers; the difference was not statistically significant (*p* = 695). Among carriers with CAP (n = 5), colonizing serotypes from PCVs were 25% from PCV10 (n = 1/4), 1.7% from PCV13 (n = 1/58), and 3.6% non-PCV10/PCV13 (n = 3/83); the difference was statistically significant (*p* = 0.047) (Table 3).

When stratified by age, pneumonia incidence was higher in the 18–23 months of age group (n = 3), in which the PCV10 colonizing serotype, 6B, had 100% (n = 1/1) of occurrence, representing a statistically significant difference (*p* = 0.016). When the presence of serotype 6B (n = 4) was compared to other serotypes (n = 141) regarding the occurrence of pneumonia (n = 5), there is an association between 6B and the pathology, responsible for 25% (n = 1/4) of the occurrences, with *p* = 0.03 (Table 4).

Table 3. Distribution of serotypes included in PCV10[a]/PCV13[b] vaccines and non-PCV10/PCV13[c] serotypes in colonized children per age group with pneumonia, Veranópolis, Brazil. Record database, n = 145.

| Age | Serotypes | Colonized children n = 145 | | Pneumonia | | |
|---|---|---|---|---|---|---|
| | | | | n | % | *p* |
| 18 to 23 months | PCV10 | 1 | 1 | 100.0 | | **0.016** |
| | PCV13 | 12 | 1 | 8.3 | | |
| | Non-PCV10/PCV13 | 14 | 1 | 7.1 | | |
| 24 to 35 months | PCV10 | 2 | 0 | 0.0 | | 0.463 |
| | PCV13 | 22 | 0 | 0.0 | | |
| | Non-PCV10/PCV13 | 16 | 1 | 6.3 | | |
| 36 to 47 months | PCV10 | 0 | 0 | 0.0 | | |
| | PCV13 | 13 | 0 | 0.0 | | |
| | Non-PCV10/PCV13 | 28 | 0 | 0.0 | | |
| 48 to 59 months | PCV10 | 1 | 0 | 0.0 | | 0.781 |
| | PCV13 | 11 | 0 | 0.0 | | |
| | Non-PCV10/PCV13 | 25 | 1 | 4.0 | | |
| Overall | PCV10 | 4 | 1 | 25.0 | | **0.047** |
| | PCV13 | 58 | 1 | 1.7 | | |
| | Non-PCV10/PCV13 | 83 | 3 | 3.6 | | |

[a]PCV10 serotypes: 1, 4, 5, 6B, 7F, 9V, 14, 18C, 19F and 23F. [b] PCV13 and non-PCV10 serotypes: 3, 6A and 19A. [c] Non-PCV10 and non-PCV13 serotypes.

Table 4. Serotypes 6B (PCV10) X Colonized children and pneumonia diagnosis, Veranópolis, Brazil. Record database, n = 145.

| Univariate | | | | | *p* | RR | CI95% | Multivariate | | |
|---|---|---|---|---|---|---|---|---|---|---|
| | Pneumonia (-) | | Pneumonia (+) | | | | | *p* | RR | CI95% |
| | n | % | n | % | | | | | | |
| 6B | 3 | 75 | 1 | 25 | **0.03** | 8.81 | 1.25-62.13 | **0.01** | 7.06 | 1.65-30.19 |
| Other serotypes[a] | 137 | 97.2 | 4 | 2.8 | | | | | | |

[a]Other serotypes: 3, 6A, 06C, 10A, 11A, 13, 15A, 15B, 15C, 16F, 16F/037, 19A, 22F, 23A, 23B, 28A, 34, 35B, NT.

## Discussion

In this cohort, we investigated the influence of *Streptococcus pneumoniae* colonization in vaccinated children regarding infectious outcomes. In general, no statistically significant differences were found in comparing the colonized group and the non-colonized group for the variables studied. Colonization by vaccine serotype 6B was rare; however, colonization by this serotype was associated to pneumonia outcome.

Our findings showed a high rate of colonization by pneumococci, over 64%, compatible with results found in other studies [11,13,14,26–28]. In Brazil, a gradual increase in the colonization rate among children has been observed in 2010, 2013, and 2017, while these rates decreased for vaccine serotypes through the years, reaching 0.9% of isolates in 2017 [11]. With the introduction of conjugate vaccines and their selective pressure, more virulent serotypes, integrated in PCV10, were eliminated [9,17]. We observed the presence of only one vaccine serotype, 6B, in 2.8% of the colonized group. Even though in lower rates, the presence of these vaccine serotypes is not negligible, due to their pathogenic potential.

Risk factors typically identified for pneumococcal colonization, such as younger age [28], attending school or having a sibling who does [9,29,30], number and age of children living in the same house [28,31], household crowding and smoke exposure [9,28,29], did not show positive association to a carrier status in our study. There are a few specificities in the studied sample that should be taken into consideration, such as the low number of children who did not attend school.

We identified association between male sex and colonization, with a 21% increase in risk (CI95% 0.99–1.48) when compared to female participants ($p = 0.05$). This was also found in two other studies [28,31], although the reasons are not clear. Few studies have been conducted about this difference usually credited to biological and behavioral factors, which also vary with age [31,32]. The question of whether the differences between sexes are caused by genetic, hormonal or environmental factors or by a combination of them can be raised. However, responding to these questions requires in-depth, multidisciplinary research. Many studies, on the other hand, did not find this difference [26,29,33].

Pneumococci evolve relatively fast in response to environmental changes, with acquired resistance to antibiotics and replacement, promoting selection and continuous expansion [28,34]. Non-vaccine serotypes (non-PCV10/13) were found in 57% of colonized children, which indicates replacement. Many Brazilian studies show that, while serotypes from PCV10 have drastically decreased although not completely disappeared, non-PCV10 serotypes have expressively increased [11,12,30], representing 57.7% of isolates in a national study in 2017 [11]. In our trial, all children were considered vaccinated; 91.1% received PCV10, and some children were vaccinated both before and after the changes to the Brazilian routine immunization schedule (from 3 + 1–2 + 1 doses), in 2016 [20]. PCV13 (3 + 1 doses) was administered in 2.7% of the children, and for 6.2%, vaccines were interchanged (mixed schedule PCV10 + PCV13). There was no association between the vaccine received and colonization. PCV10 coverage in Veranópolis reached a 98% of children (2016–2020, DATA-SUS), showing that being colonized or not, in this scenario, does not change the risk for the disease.

Among the outcomes investigated in this study, no association with colonization was observed, which emphasizes the concept of lower virulence in non-vaccine serotypes. Pneumonia was observed in only 7/225 (3.1%) of the children in this study, being 5/145 (3.4%) among colonized and 2/80 (2.5%) among non-colonized children ($p = 0.695$). Not all outcomes presented are directly related to pneumococci. However, taking into account that it can occur through interaction and co-infection by many potential pathogens (viral and bacterial) from the host's NP [29], this relation should be considered. Petraitiene *et al*., 2016, observed that pneumococcal colonization is associated to higher pneumonia, sinusitis, and AOM frequencies and to antimicrobial treatment during the episode in comparison to non-colonized children [29]. On the other hand, our findings show that children colonized by pneumococcus do not have an increased risk for respiratory diseases or antimicrobial use. Data have not varied among colonized and non-colonized children, as can be seen in the overall mean of visits (3.4), mean of visits per season (1.1 visit in spring), antimicrobial prescription (average 51.6%), among others. Although this study has not been developed to evaluate vaccination effectiveness, it is reasonable to consider the effect of this intervention on the incidence of infections.

We found association between serotype 6B colonization and pneumonia through the one-year period, especially in children under 2 years ($p = 0.016$). Although it is not possible to establish that the colonizing pneumococcus caused CAP

due to the lack of a documented etiology and the limitations of the study design, the association between the capsular type of the vaccine and this CAP cannot be neglected. Pneumococcus is recognized as the most important agent of bacterial pneumonia [3,13,16,23,35]. Many simulations were conducted between serotype 6B colonization and its relation to pneumonia in the studied cohort, and all resulted in significant association, even with few observed cases. In a vaccinated population, SIREVA network found 4.7% of samples belonging to serotype 6B [36], which shows the continuity of its circulation in post-PCV10 era.

There is substantial variation between individuals in the immune response to vaccination and this occurrence may be linked to factors: intrinsic, extrinsic, environmental, behavioral and nutritional [37,38]. But three factors must be taken into consideration: first, serotype 6B is one of the least effective components of PCV(82.8%) [38,39]; second, there is a reduction in the serospecific response when colonization (6B) occurs before the administration of the first dose of PCV [21,38,40]; third, natural immunity may only be achieved after many episodes of colonization or infection [38,40]. An understanding of all these factors and their impacts offers ways to improve vaccine immunogenicity and efficacy [37]. Association between serotype 6B colonization and pneumonia occurrence in a one-year period is an unprecedented observation, with no similar account found to our knowledge.

This study presents some limitations, including the study design, which prevents us from defining causality. It is also important to mention the lack of standardized definition of criteria for diagnoses among pediatricians and memory bias in interviews with the guardians (minimized by information verified against medical records). The interruption of school activities imposed by social isolation during the SARS-CoV-2 pandemic may have influenced children's exposure to pathogens and the incidence of respiratory infections; however, while the impact of social isolation on *S. pneumoniae* colonization remains underexplored, studies have observed circulation of pneumococci belonging to NVT among patients suspected of having COVID-19 during the pandemic's first year [41,42]. On the other hand, our study presents strengths, such as prospective follow-up of a substantial number of infants to preschoolers with longitudinal follow-up for a one-year period.

Although pneumococcal colonization in children has not been shown to increase the risk of respiratory diseases, the residual circulation of vaccine serotypes, such as 6B, warrants attention due to its association with pneumonia. This concern remains valid despite the reduced number of cases observed in this study, as the role of pneumococcus as the leading etiological agent of bacterial pneumonia in children cannot be underestimated. While this study did not directly evaluate vaccine effectiveness, the high vaccination coverage observed suggests that colonization in this context does not significantly alter the risk of disease, highlighting the impact of vaccines. However, it is crucial to monitor serotype replacement and the growing diversity of pneumococcal strains, which could undermine the benefits achieved. The findings reinforce the effectiveness of the strategies implemented by the National Immunization Program (PNI) and underscore the importance of preserving these achievements. Continuous strain analysis, epidemiological surveillance, and future studies are essential to guide adaptations in immunization policies, particularly regarding residual serotypes like 6B, which was the second most common in the pediatric population during the pre-PCV era and associated with severe infections. These efforts are vital to ensuring effective protection against pneumococcal disease, considering that, so far, vaccination is the only available tool.

## Acknowledgments

• Centers for Disease Control and Prevention (CDC) for performing pneumococcal serotyping.
• Moriguchi Institute for helping us with secretariat support and logistics.
• Cristiane Bündchen for providing statistical assistance.
• Veranópolis Health Department and HCSPL (Hospital São Peregrino Lazziozi) for collecting information from medical records.
• We thank all children and their parents/guardians who took part in the study and the nurses, pediatricians and community health workers at the Public and Private Health Systems of Veranópolis.

## Author contributions

**Conceptualization:** Jaqueline Elisa Verardo Benedetti, Mariana Preussler Mott, Emilio Hideyuki Moriguchi, Roberta Rigo Dalla Corte, João Carlos Batista Santana, Cícero Dias.

**Data curation:** Jaqueline Elisa Verardo Benedetti, Kauana Pizzutti, Mariana Preussler Mott, Pedro Uriel Pedrotti Vieira, Cícero Dias.

**Formal analysis:** Jaqueline Elisa Verardo Benedetti, Mariana Preussler Mott, João Carlos Batista Santana, Cícero Dias.

**Funding acquisition:** Emilio Hideyuki Moriguchi, Cícero Dias.

**Investigation:** Jaqueline Elisa Verardo Benedetti, Kauana Pizzutti, Mariana Preussler Mott, Pedro Uriel Pedrotti Vieira, Neide Maria Bruscato, Cícero Dias.

**Methodology:** Jaqueline Elisa Verardo Benedetti, Kauana Pizzutti, Mariana Preussler Mott, Roberta Rigo Dalla Corte, João Carlos Batista Santana, Cícero Dias.

**Project administration:** Jaqueline Elisa Verardo Benedetti, Neide Maria Bruscato, Emilio Hideyuki Moriguchi, Cícero Dias.

**Resources:** Neide Maria Bruscato, Emilio Hideyuki Moriguchi, Cícero Dias.

**Software:** Jaqueline Elisa Verardo Benedetti, Pedro Uriel Pedrotti Vieira, Cícero Dias.

**Supervision:** Jaqueline Elisa Verardo Benedetti, Emilio Hideyuki Moriguchi, João Carlos Batista Santana, Cícero Dias.

**Validation:** Jaqueline Elisa Verardo Benedetti, Kauana Pizzutti, Mariana Preussler Mott, Pedro Uriel Pedrotti Vieira, João Carlos Batista Santana, Cícero Dias.

**Visualization:** Jaqueline Elisa Verardo Benedetti, João Carlos Batista Santana, Cícero Dias.

**Writing – original draft:** Jaqueline Elisa Verardo Benedetti, João Carlos Batista Santana, Cícero Dias.

**Writing – review & editing:** Jaqueline Elisa Verardo Benedetti, Kauana Pizzutti, Mariana Preussler Mott, Neide Maria Bruscato, Emilio Hideyuki Moriguchi, Roberta Rigo Dalla Corte, João Carlos Batista Santana, Cícero Dias.

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
