## [Decision Letter · Decision Letter 0]

Dear Dr. Dias,

Your study is quite interesting, however, you should consider some aspects such as: improve the wording of the summary section, review the sample size calculation in the sample section, and the power of the study. You should also strengthen the discussion of the research, and clearly specify the implications of the research. 

Please submit your revised manuscript by Nov 29 2024 11:59PM. If you will need more time than this to complete your revisions, please reply to this message or contact the journal office at plosone@plos.org . A rebuttal letter that responds to each point raised by the academic editor and reviewer(s). You should upload this letter as a separate file labeled 'Response to Reviewers'.A marked-up copy of your manuscript that highlights changes made to the original version. You should upload this as a separate file labeled 'Revised Manuscript with Track Changes'.An unmarked version of your revised paper without tracked changes. You should upload this as a separate file labeled 'Manuscript'.

We look forward to receiving your revised manuscript.

Kind regards,

Oriana Rivera-Lozada de Bonilla

Academic Editor

PLOS ONE

2. In the online submission form, you indicated that [All relevant data are within the manuscript. Database available on request to the author.]. All PLOS journals now require all data underlying the findings described in their manuscript to be freely available to other researchers, either 1. In a public repository, 2. Within the manuscript itself, or 3. Uploaded as supplementary information. This policy applies to all data except where public deposition would breach compliance with the protocol approved by your research ethics board. If your data cannot be made publicly available for ethical or legal reasons (e.g., public availability would compromise patient privacy), please explain your reasons on resubmission and your exemption request will be escalated for approval.

1. Is the manuscript technically sound, and do the data support the conclusions?

Reviewer #1: Partly

Reviewer #2: Yes

2. Has the statistical analysis been performed appropriately and rigorously?

Reviewer #1: Yes

Reviewer #2: Yes

3. Have the authors made all data underlying the findings in their manuscript fully available?

Reviewer #1: Yes

Reviewer #2: No

4. Is the manuscript presented in an intelligible fashion and written in standard English?

Reviewer #1: Yes

Reviewer #2: No

Reviewer #1: This study aims to explore the relationship between pneumococcal vaccination, Streptococcus pneumoniae colonization, and potential respiratory complications following vaccination. While the findings offer valuable insights, certain issues related to sample selection, and methodology may influence the reliability and generalizability of the results.

Line 1: The title could be made more concise and precise: Post-Vaccination Streptococcus pneumoniae Colonization and Respiratory Manifestations in Children: A Prospective Cohort Study

Line 35: The abstract should be organized into distinct subsections: background, methods, results, and conclusion.

Line 43: In the abstract section, it is sufficient to mention the number of participants who successfully completed the follow-up, without specifying the number of individuals who dropped out of the study.

Line 61-67: The authors have effectively addressed the impact of the vaccination program on reducing mortality associated with pneumococcal respiratory infections. it would be beneficial to provide data on the incidence rates of both invasive and non-invasive pneumococcal disease among children in the study area, Brazil, or similar regions, both before and after the introduction of the vaccination program. Clarification is needed regarding whether the immunity conferred by the PCV10 and PCV13 vaccines primarily prevents the occurrence of infections or mitigates the severity of the disease.

Line 78: The author should clarify the statement, “post-conjugate vaccine introduction studies did not find a decrease in overall carriage rates in the population, but instead observed a simultaneous replacement by NVT." It should be noted that the introduction of conjugate vaccines led to serotype replacement, where non-vaccine serotypes (NVTs) became more prevalent within the population.

The following references are well-established in the literature to support these findings:

Hammitt, L. L., et al. (2014). "Effect of pneumococcal conjugate vaccine introduction on nasopharyngeal pneumococcal carriage in Kenyan children: a cross-sectional study." The Lancet Global Health.

Weinberger, D. M., et al. (2011). "Serotype replacement in disease after pneumococcal vaccination." The Lancet.

van Hoek, A. J., et al. (2014). "The effect of pneumococcal conjugate vaccine on the carriage of Streptococcus pneumoniae in England: a cross-sectional study." PLOS Medicine.

Line 110: The study's recruitment strategy, which relies heavily on one or two local schools, may introduce selection bias by potentially excluding non-school-attending children or students from other schools in different urban or rural areas. Clarify any potential biases resulting from the recruitment method. The study also heavily depends on data from prior research, limiting the capacity to control for all variables.

Line 128: Relying on non-standardized diagnostic criteria used by physicians for respiratory diseases may introduce variability and reduce diagnostic consistency. The authors should address the implications of using non-standardized diagnostic criteria across different physicians and its potential impact on the consistency of outcome measures. Could this variability have biased the results, leading to an underestimation or overestimation of disease incidence?

Line 143: The outcomes, particularly pneumonia, were rigorously defined using criteria from Shah et al., enhancing the study's internal validity. However, for pneumonia specifically, while chest X-rays were used to confirm cases, it remains unclear whether all children with suspected pneumonia underwent imaging. This potential inconsistency may have introduced bias. The authors should discuss whether all suspected pneumonia cases were subjected to chest X-ray confirmation, and how missing data or inconsistent confirmation might have influenced the study’s findings.

Line 146: Nasopharyngeal (NP) swabs were collected only at the initial time point, which may not fully capture the dynamics of colonization over time. A longitudinal sampling approach, involving multiple NP swabs over time, could have provided more comprehensive data on colonization persistence or clearance.

Line 160: The sample size calculation appears well-justified; however, details regarding the actual statistical power based on the final dataset are missing. A more thorough discussion is needed to determine whether the achieved sample size was sufficient to detect the hypothesized effect size, particularly within subgroups. Additionally, consider addressing potential confounding factors in the Poisson regression analysis, such as socioeconomic status, healthcare access, or pre-existing health conditions.

Line 290: The text contains several small formatting errors, such as redundant citations and minor repetition (e.g., "24, 24" in multiple places).

Line 307: The implications of the study for public health and vaccine policy are not emphasized enough, despite the study having potential relevance for the National Immunization Program in Brazil. A more explicit connection between the study's findings and how they could inform vaccine strategies or policies would be valuable. For example, the implications of continued circulation of serotype 6B should be further discussed in the context of vaccine development or the need for updated vaccines.

Line 315: The lack of association between colonization and respiratory outcomes is an important finding, yet the discussion does not explore potential reasons why this might be the case. The authors could elaborate on possible explanations, such as the effectiveness of the vaccines in preventing disease despite colonization, or the role of other pathogens and co-infections in respiratory disease outcomes. This would provide a more thorough interpretation of the results.

Line 350: The authors briefly mention the impact of the COVID-19 pandemic on the study, but this section could be expanded.

Reviewer #2: REVIEWER COMMENTS

General

Verardo et al assessed the effect of S. pneumoniae colonization on risk of respiratory diseases among children aged 18-59 months who are fully vaccinated with PCV. The study is important and could contribute to knowledge in the area. Generally, it is well written but would require proofreading to fix minor language errors, as well as clarification of some technical issues.

Abstract

1. Lines 48-49” and the presence of only one PCV10 vaccine serotype, 6B (2.8% of colonized)”: Was it expected that more vaccine serotypes will be colonized among the study participants who have been vaccinated? Please, clarify

2. Lines 51-52 “Restriction was only observed in cases of serotype 6B colonization….”: This statement is not clear. Does “restriction” mean “exception”? Please reword

Introduction

1. Lines 63-64: What accounted for the decreased mortality between 2000 and 2015?

2. Lines 65-67: Within which period was the 88% reduction? What about the 51% reduction? These statements appear confusing, kindly reconcile

Methods and material

1. Study design

a. It is not clear which period the study was undertaken

b. Lines 93-100: Separate the study design from the profile of the study areas. Improve the profile of the study area by include access to healthcare i.e. number of healthcare facilities; top causes of morbidity and mortality among children 18-59 months, the position of S. pneumoniae-associated respiratory infections on the log of top causes of morbidity and mortality; map of the study area etc.

2. Recruitment

a. Line 103: Consider substituting “10-valent conjugate vaccine” with PCV10 for consistency

3. Follow-up

a. Line 119: Change “closed questions” to “close-ended questions”

b. Line 123: What is the difference between “SimPCV13” and PCV13? Please ensure consistency

Results

1. Line 185: The period of follow up (March/2019 to October/2020) is more than one year. Kindly reconcile this with the respective sections e.g. title, line 186 etc.

2. Line 190: The text in Fig.1 is blur. Kindly replace with a better image

3. Line 197: Change “colonized” to “colonization”

4. Lines 231 “In 51% of medical care visits for both carriers and non-carriers of S. pneumoniae, there was antimicrobial prescription (51.7%, n=75/145 versus 51.3%, n=41/80)”. This statement is not clear. 51.7% vrs 51.3% do not round up to 51%, respectively. Kindly clarify

5. Table 2: What was the comparison group for the respective variables?

Discussion

1. Line 276 “no differences”: The were differences, except that they were not statistically significant. Kindly reconcile

2. Line 340 “hyporesponsiveness due to vaccine failure”. Vaccine failure may be inherent to the vaccine, and hypo-responsiveness may be person-related outcome. Hypo-responsiveness may not be as a result of the vaccine but biological factors of the individual including genetic and nutritional. Please revise the statement

**Do you want your identity to be public for this peer review?** For information about this choice, including consent withdrawal, please see our Privacy Policy

Reviewer #1: No

Reviewer #2: **Yes: ** Michael Rockson Adjei

---

## [Author Response · Author response to Decision Letter 1]

21 Mar 2025

Reviewer 1

The specific comments:

Reviewer #1: This study aims to explore the relationship between pneumococcal vaccination, Streptococcus pneumoniae colonization, and potential respiratory complications following vaccination. While the findings offer valuable insights, certain issues related to sample selection, and methodology may influence the reliability and generalizability of the results.

Your notes were very important. Each one was answered as follows below.

Line 1: The title could be made more concise and precise: Post-Vaccination Streptococcus pneumoniae Colonization and Respiratory Manifestations in Children: A Prospective Cohort Study.

Thank you for the suggestion about the title. We accept the reviewer suggestion and the manuscript has a different title: “Post-Vaccination Streptococcus pneumoniae Colonization and Respiratory Manifestations in Children: A Prospective Cohort Study”

Line 35: The abstract should be organized into distinct subsections: background, methods, results, and conclusion.

We appreciate the suggestion, and the summary section now contains distinct subsections: Background, Methods, Results, Conclusion. It was modified as suggested.

Line 43: In the abstract section, it is sufficient to mention the number of participants who successfully completed the follow-up, without specifying the number of individuals who dropped out of the study.

It was modified as suggested.

“A total of 225 children were monitored, with different dates of entry into the study, which occurred between March 2018 and October 2019 (zero time)."

Line 61-67: The authors have effectively addressed the impact of the vaccination program on reducing mortality associated with pneumococcal respiratory infections. it would be beneficial to provide data on the incidence rates of both invasive and non-invasive pneumococcal disease among children in the study area, Brazil, or similar regions, both before and after the introduction of the vaccination program. Clarification is needed regarding whether the immunity conferred by the PCV10 and PCV13 vaccines primarily prevents the occurrence of infections or mitigates the severity of the disease.

Thank you for the excellent suggestion. Unfortunately, in Brazil, there is no surveillance for pneumonia or IPD in all regions of the country; therefore, there are no study data available that have clearly demonstrated a decrease in the incidence of IPD in the Southern region (site of the study in question). We have included bibliographic references as per the text below:

“In their study, Wahl et al. (2018) estimated that there were 294,000 pneumococcal deaths among children aged 1–59 months worldwide in 2015 and that these deaths decreased by 51% from 2000 to 2015 due to the widespread use of PCV accompanied by also a reduction in cases of the disease 4,5. The same study highlighted that Brazil reached one of the largest reductions by 2015, 88% of deaths from this agent after the introduction of the 10-valent pneumococcal conjugate vaccine (PCV10) in the country4. More recently, in a hospital-based study historical series on IPD among children in Brazil, fatalities were significantly reduced from 6.6 to 2.0 cases per 10,000 after the introduction of PCV10 in the public system6. In the state of Rio Grande do Sul, a decrease in the incidence of meningitis has been observed over time7.

Bibliographic references:

Bardach, A., Ruvinsky, S., Palermo, M. C., Alconada, T., Sandoval, M. M., Brizuela, M. E., Wierzbicki, E. R., Cantos, J., Gagetti, P., & Ciapponi, A. (2024). Invasive pneumococcal disease in Latin America and the Caribbean: Serotype distribution, disease burden, and impact of vaccination. A systematic review and meta-analysis. PLoS ONE, 19(6 June). https://doi.org/10.1371/journal.pone.0304978

Berezin, E. N., Jarovsky, D., Cardoso, M. R. A., & Mantese, O. C. (2020). Invasive pneumococcal disease among hospitalized children in Brazil before and after the introduction of a pneumococcal conjugate vaccine. Vaccine, 38(7), 1740–1745. https://doi.org/10.1016/J.VACCINE.2019.12.038

Centro Estadual de Vigilância em Saúde. (2023). INFORME EPIDMIOLÓGICO DAS MENINGITES 2018-2022.

Line 78: The author should clarify the statement, “post-conjugate vaccine introduction studies did not find a decrease in overall carriage rates in the population, but instead observed a simultaneous replacement by NVT." It should be noted that the introduction of conjugate vaccines led to serotype replacement, where non-vaccine serotypes (NVTs) became more prevalent within the population.

Thank you for suggestion. We have included suggested bibliographic references and changed the text as follows.:

“Post-conjugate vaccine introduction studies have not found a decrease in overall carriage rates in the population, but rather replacement, where non-vaccine serotypes (NVT) have become more prevalent13-15.

Bibliographic reference:

Weinberger, D. M., Malley, R., & Lipsitch, M. (2011). Serotype replacement in disease after pneumococcal vaccination. The Lancet, 378(9807), 1962–1973. https://doi.org/10.1016/S0140-6736(10)62225-8/ASSET/F7618DAA-6766-491D-8BE2-40B062953AB9/MAIN.ASSETS/GR1.SML

Line 110: The study's recruitment strategy, which relies heavily on one or two local schools, may introduce selection bias by potentially excluding non-school-attending children or students from other schools in different urban or rural areas. Clarify any potential biases resulting from the recruitment method. The study also heavily depends on data from prior research, limiting the capacity to control for all variables.

We agree with your comment, as the text described suggested that only 2 schools were accessed, but in fact the process involved 11 out of 13 schools in the community.

It was modified as suggested line 121 to 125: “Following the random draw order, 11 out of 13 school principals were contacted to ensure a randomized selection. School visits continued until the required sample size was reached, considering a loss of 10%, resulting in different entry and exit dates for each participant. Additional volunteers outside the school environment were recruited through an appeal on local radio.”

Line 128: Relying on non-standardized diagnostic criteria used by physicians for respiratory diseases may introduce variability and reduce diagnostic consistency. The authors should address the implications of using non-standardized diagnostic criteria across different physicians and its potential impact on the consistency of outcome measures. Could this variability have biased the results, leading to an underestimation or overestimation of disease incidence?

We agree with your comment that there really is a potential bias in relation to the diagnosis because it is a variable that is not fully controlled and this appears recorded in the study limitations section between lines 357 to 364 (text below). However, if on the one hand there is a potential bias, on the other hand all diagnoses were rigorously evaluated and confirmed/ruled out by a pediatrician on our team with extensive clinical and academic experience in respiratory diseases.

“It is also important to mention the lack of standardized definition of criteria for diagnoses among pediatricians and memory bias in interviews with the guardians (minimized by information verified against medical records).”

Line 143: The outcomes, particularly pneumonia, were rigorously defined using criteria from Shah et al., enhancing the study's internal validity. However, for pneumonia specifically, while chest X-rays were used to confirm cases, it remains unclear whether all children with suspected pneumonia underwent imaging. This potential inconsistency may have introduced bias. The authors should discuss whether all suspected pneumonia cases were subjected to chest X-ray confirmation, and how missing data or inconsistent confirmation might have influenced the study’s findings.

We agree with your observation that we do not have absolute control over this issue. However, in the community where the study was conducted, children often undergo radiologic evaluation when pneumonia is suspected in the pediatric clinic. That said, we cannot guarantee that all suspected cases underwent radiologic investigation. We emphasize that all diagnoses were rigorously evaluated and confirmed/ruled out by a pediatrician on our staff with solid expertise in respiratory diseases, as described in line 153 to 155 of the manuscript: "All pneumonia cases were reviewed and confirmed/ruled by our team's pediatrician (JCBS), considering the previously described criteria for case entry together with the demonstration of infiltrate or consolidation on chest X-ray”. However, this limitation is addressed in the study discussion section, line 357 to 364: “It is also important to mention the lack of standardized definition of criteria for diagnoses among pediatricians….”.

Line 146: Nasopharyngeal (NP) swabs were collected only at the initial time point, which may not fully capture the dynamics of colonization over time. A longitudinal sampling approach, involving multiple NP swabs over time, could have provided more comprehensive data on colonization persistence or clearance.

We agree with your comment that ideally monitoring involving multiple serial NP swabs would have provided greater consistency to the study, but it was not designed to monitor colonization over time.

Line 160: The sample size calculation appears well-justified; however, details regarding the actual statistical power based on the final dataset are missing. A more thorough discussion is needed to determine whether the achieved sample size was sufficient to detect the hypothesized effect size, particularly within subgroups. Additionally, consider addressing potential confounding factors in the Poisson regression analysis, such as socioeconomic status, healthcare access, or pre-existing health conditions.

In fact, we reconsidered some issues, we appreciate the suggestion and, as shown in tables 1, 2 and 4, we included the multivariate analysis adjusted for age and sex with the intention of clarifying possible confounding factors.

Table 1 - Cohort of children aged 18–59 months, colonized and non-colonized: general (sociodemographic) and vaccination characteristics, Veranópolis, Brazil — Interview database, n=225.

Univariate Multivariate

VARIABLE S. pneumoniae (-) S. pneumoniae (+) p RR CI95% p RR CI95%

n=80 n=145

n (%) n (%)

Age: 18 to 23 months 16 (37.2%) 27 (62.8%) 1 1

24 to 35 months 23 (36.5%) 40 (63.5%) 0,941 1,011 0,752 1,36 0,840 1,030 0,773-1,372

36 to 47 months 21 (33.9%) 41 (66.1%) 0,727 1,053 0,787 1,409 0,522 1,104 0,815-1,497

48 to 59 months 20 (35.1%) 37 (64.9%) 0,828 1,034 0,767 1,394 0,463 1,118 0,830-1,506

Sex: Female 46 (41.8%) 64 (58.2%) 1 1

Male 34 (29.6%) 81 (70.4%) 0,05 1,21 0,99 1,48 0,117 1,177 0,960-1,443

Householda: 2–3 people 33 (35.1%) 61 (64.9%) 1 1

4–5 people 40 (34.8%) 75 (65.2%) 0,961 1,005 0,823 1,227 0,585 1,061 0,857-1,314

6 or more 7 (43.8%) 9 (56.3%) 0,540 0,867 0,549 1,369 0,404 0,826 0,526-1,296

Children in the householdb: 0-1 64 (33.9%) 125 (66.1%) 1 1

2–3 16 (44.4%) 20 (55.6%) 0,269 0,84 0,616 1,145 0,194 0,802 0,574-1,119

Shared bedroomc: No 45 (34.1%) 87 (65.9%) 1 1

Yes 35 (37.6%) 58 (62.4%) 0,588 0,946 0,775 1,156 0,363 0,906 0,732-1,121

Smoker family member: No 64 (35%) 119 (65%) 1 1

Yes 16 (38.1%) 26 (61.9%) 0,703 0,952 0,734 1,235 0,529 0,924 0,723-1,181

Smoke exposured: No 36 (37.1%) 61 (62.9%) 1 1

Yes 44 (34.4%) 84 (65.6%) 0,671 1,044 0,856 1,272 0,870 1,017 0,834-1,239

Schoole: No 2 (40%) 3 (60%) 1 1

Yes 78 (35.5%) 142 (64.5%) 0,843 1,076 0,522 2,215 0,729 1,128 0,570-2,234

Vaccination: Mixedf 6 (42.9%) 8 (57.1%) 1 1

PCV13g 3 (50%) 3 (50%) 0,776 0,875 0,349 2,195 0,796 0,887 0,358-2,199

PCV10h 71 (34.6%) 134 (65.4%) 0,570 1,144 0,719 1,820 0,599 1,132 0,713-1,796

S. pneumoniae (-) = non-colonized, S. pneumoniae (+) = colonized. a Number of people living in the same household (including the child). b Number of children under 5 in the household (including the child). c Child shares the bedroom or bed with at least one person. d Use of fireplace or wood cookstove in the house. e School attendance between “zero time” and “end of follow-up”. f Mixed vaccination — the child received PCV10 and PCV13 doses. g Received 4 doses of PCV13. h Received 3 or 4 doses of PCV10.

Table 2 - Distribution of outcomes per clinic syndromea among children aged 18–59 months with medical care in record, Veranópolis, Brazil, n=225.

Univariate Multivariate

S.pneumoniae (-) n=80 S.pneumoniae (+) n=145 p RR IC95% p RR IC95%

n % n %

Pneumonia 2 2,50% 5 3,40% 0,695 1,38 0,27 6,95 0,714 1,37 0,25 7,36

Sinusitis 14 17,50% 25 17,20% 0,961 0,99 0,54 1,79 0,985 1,01 0,55 1,84

AOMb 14 17,50% 26 17,90% 0,935 1,02 0,57 1,85 0,850 1,06 0,59 1,90

Tonsillopharyngitis 22 27,50% 36 24,80% 0,661 0,9 0,57 1,42 0,548 0,87 0,55 1,37

ARIc 66 82,50% 109 75,20% 0,206 0,91 0,79 1,05 0,382 0,94 0,82 1,08

Asthma 2 2,50% 9 6,20% 0,217 2,48 0,55 11,2 0,319 2,22 0,46 10,72

Antimicrobiald 41 51,30% 75 51,70% 0,946 1,01 0,77 1,32 0,972 1,00 0,77 1,31

No medical caree 8 10% 29 20% 0,064 2,00 0,96 4,17 0,096 1,82 0,90 3,68

a Presence of at least one episode of respiratory disease (pneumonia, sinusitis, AOM, ARI, asthma or tonsillopharyngitis) between “zero time” and “end of follow-up”. b AOM — acute otitis media. c ARI — Upper and lower respiratory tract infection — diagnosis classified in this category when record showed: cold, flu, rhinitis, allergic/viral rhinitis, respiratory virosis, conjunctivitis, influenza, viral tonsillitis, sore throat, nasopharyngitis, pharyngitis, laryngitis, turbinate hypertrophy, enlarged adenoids, tracheobronchitis, bronchitis, bronchiolitis, acute respiratory infection (ARI), head, foot, and mouth disease, erythema multiform, mononucleosis, adenovirus infection, unspecified viral infection, viral/external/serous AOM, positive COVID test. d Record showed one or more uses of antimicrobial between “zero time” and “end of follow-up”. e Children without medical care record in the period of interest.

Table 4 - Serotypes 6B (PCV10) X Colonized children and pneumonia diagnosis, Veranópolis, Brazil. Record database, n=145.

Univariate Multivariate

Pneumonia (-) Pneumonia (+) p RR CI95% p RR CI95%

n % n %

6B 3 75,0% 1 25% 0,03 8,81 1,25 62,1 0,01 7,06 1,65 30,19

Other serotypesa 137 97,2% 4 2,80%

aOther serotypes: 3, 6A, 06C, 10A, 11A, 13, 15A, 15B, 15C, 16F, 16F/037, 19A, 22F, 23A, 23B, 28A, 34, 35B, NT.

Line 290: The text contains several small formatting errors, such as redundant citations and minor repetition (e.g., "24, 24" in multiple places).

It was modified as suggested.

Line 307: The implications of the study for public health and vaccine policy are not emphasized enough, despite the study having potential relevance for the National Immunization Program in Brazil. A more explicit connection between the study's findings and how they could inform vaccine strategies or policies would be valuable. For example, the implications of continued circulation of serotype 6B should be further discussed in the context of vaccine development or the need for updated vaccines.

It was modified as suggested between lines 367 to 382.

“Although pneumococcal colonization in children has not been shown to increase the risk of respiratory diseases, the residual circulation of vaccine serotypes, such as 6B, warrants attention due to its association with pneumonia. This concern remains valid despite the reduced number of cases

---

## [Decision Letter · Decision Letter 1]

Post-Vaccination Streptococcus pneumoniae Colonization and Respiratory Manifestations in Children: A Prospective Cohort Study

PONE-D-24-16432R1

Dear Dr. Cícero Dias,

We’re pleased to inform you that your manuscript has been judged scientifically suitable for publication and will be formally accepted for publication once it meets all outstanding technical requirements.

Kind regards,

Oriana Rivera-Lozada de Bonilla

Academic Editor

PLOS ONE

**Comments to the Author**

Reviewer #3: (No Response)

2. Is the manuscript technically sound, and do the data support the conclusions?

Reviewer #3: Yes

3. Has the statistical analysis been performed appropriately and rigorously?

Reviewer #3: Yes

4. Have the authors made all data underlying the findings in their manuscript fully available?

Reviewer #3: Yes

5. Is the manuscript presented in an intelligible fashion and written in standard English?

Reviewer #3: Yes

Reviewer #3: The manuscript presents a prospective cohort study examining the relationship between Streptococcus pneumoniae colonization and respiratory disease outcomes in a vaccinated pediatric population in southern Brazil. The study is timely and relevant. The authors are to be commended for their extensive and thoughtful responses to the two previous reviews. The manuscript is well organized and substantially improved. However, some critical issues remain regarding exposure assessment, the interpretation of key findings, and the framing of conclusions. These do not invalidate the work but warrant additional clarification

• The study classifies children as "colonized" or "non-colonized" based on a single nasopharyngeal swab at baseline (“zero time”). While this limitation is acknowledged, it significantly affects the ability to draw conclusions over a one-year follow-up period. Colonization is known to be dynamic in this age group.

The discussion should more explicitly consider the implications of this limitation, including possible exposure misclassification and its likely direction of bias.

• The reported association between colonization with serotype 6B and pneumonia appears to stem from a single case among four colonized children. Although this yields a statistically significant relative risk, the absolute number is too small to draw robust inferences.

Reframe this association as exploratory and hypothesis-generating.

• The finding of no significant association between overall colonization and respiratory illness is important. However, alternative explanations should be more fully explored in the discussion—for example: effective immunity, low pathogenicity of the prevailing non-vaccine serotypes, possible diagnostic under-reporting due to mild symptoms or pandemic disruptions. Expand on these potential explanations.

**Do you want your identity to be public for this peer review?** For information about this choice, including consent withdrawal, please see our Privacy Policy

Reviewer #3: No

---

## [Editor Report · Acceptance letter]

PONE-D-24-16432R1

PLOS ONE

Dear Dr. Dias,

I'm pleased to inform you that your manuscript has been deemed suitable for publication in PLOS ONE. Congratulations! Your manuscript is now being handed over to our production team.

Kind regards,

on behalf of

Dr. Oriana Rivera-Lozada de Bonilla

Academic Editor

PLOS ONE